# Serum NfL in Alzheimer Dementia: Results of the Prospective Dementia Registry Austria

**DOI:** 10.3390/medicina58030433

**Published:** 2022-03-16

**Authors:** Daniela Kern, Michael Khalil, Lukas Pirpamer, Arabella Buchmann, Edith Hofer, Peter Dal-Bianco, Elisabeth Stögmann, Christoph Scherfler, Thomas Benke, Gerhard Ransmayr, Reinhold Schmidt

**Affiliations:** 1Department of Neurology, Medical University of Graz, Auenbruggerplatz 22, 8036 Graz, Austria; daniela.eibl@medunigraz.at (D.K.); michael.khalil@medunigraz.at (M.K.); lukas.pirpamer@medunigraz.at (L.P.); arabella.buchmann@medunigraz.at (A.B.); edith.hofer@medunigraz.at (E.H.); 2Institute for Medical Informatics, Statistics and Documentation, Medical University of Graz, Auenbruggerplatz 2, 8036 Graz, Austria; 3Department of Neurology, Medical University of Vienna, Währinger Gürtel 18-20, 1090 Vienna, Austria; peter@dal-bianco.at (P.D.-B.); elisabeth.stoegmann@meduniwien.ac.at (E.S.); 4Department of Neurology, Innsbruck Medical University, Anichstraße 35, 6020 Innsbruck, Austria; christoph.scherfler@i-med.ac.at (C.S.); thomas.benke@i-med.ac.at (T.B.); 5Department of Neurology 2, Faculty of Medicine, Kepler University Hospital, Krankenhausstraße 9, 4020 Linz, Austria; gerhard.ransmayr@kepleruniklinikum.at

**Keywords:** Alzheimer’s, neurofilament light (NfL), biomarker, blood biomarker, dementia

## Abstract

*Background and Objectives*: The neurofilament light chain (NfL) is a biomarker for neuro-axonal injury in various acute and chronic neurological disorders, including Alzheimer’s disease (AD). We here investigated the cross-sectional and longitudinal associations between baseline serum NfL (sNfL) levels and cognitive, behavioural as well as MR volumetric findings in the Prospective Dementia Registry Austria (PRODEM-Austria). *Materials and Methods*: All participants were clinically diagnosed with AD according to NINCDS-ADRDA criteria and underwent a detailed clinical assessment, cognitive testing (including the Mini Mental State Examination (MMSE) and the Consortium to Establish a Registry for Alzheimer’s Disease (CERAD)), the neuropsychiatric inventory (NPI) and laboratory evaluation. A total of 237 patients were included in the study. Follow-up examinations were done at 6 months, 1 year and 2 years with 93.3% of patients undergoing at least one follow-up. We quantified sNfL by a single molecule array (Simoa). In a subgroup of 125 subjects, brain imaging data (1.5 or 3T MRI, with 1 mm isotropic resolution) were available. Brain volumetry was assessed using the FreeSurfer image analysis suite (v6.0). *Results*: Higher sNfL concentrations were associated with worse performance in cognitive tests at baseline, including CERAD (B = −10.084, SE = 2.999, *p* < 0.001) and MMSE (B = −3.014, SE = 1.293, *p* = 0.021). The sNfL levels also correlated with the presence of neuropsychiatric symptoms (NPI total score: r = 0.138, *p* = 0.041) and with smaller volumes of the temporal lobe (B = −0.012, SE = 0.003, *p* = 0.001), the hippocampus (B = −0.001, SE = 0.000201, *p* = 0.013), the entorhinal (B = −0.000308, SE = 0.000124, *p* = 0.014), and the parahippocampal cortex (B = −0.000316, SE = 0.000113, *p* = 0.006). The sNfL values predicted more pronounced cognitive decline over the mean follow-up period of 22 months, but there were no significant associations with respect to change in neuropsychiatric symptoms and brain volumetric measures. *Conclusions*: the sNfL levels relate to cognitive, behavioural, and imaging hallmarks of AD and predicts short term cognitive decline.

## 1. Introduction

Neurofilament light chain (NfL) is a cytoskeletal protein exclusively expressed in neurons. After axonal damage, NfL is released consecutively into the extracellular fluid, the cerebrospinal fluid (CSF) and the blood [1]. Using ultrasensitive single molecule array assays (Simoa) it is possible to measure low concentrated NfL, not only in CSF but also in blood samples, with very high sensitivity. Blood levels correlate strongly to those measured in CSF [2,3,4].

Elevated NfL levels in CSF and peripheral blood have been described in a number of neurological conditions including Alzheimer’s disease (AD). According to Bridel et al., the highest NfL levels are seen in HIV-associated dementia, frontotemporal dementia, vascular dementia, amyotrophic lateral sclerosis and atypical parkinsonian syndromes [5]. However, because of its low specificity for AD, NfL is not suitable for the differentiation between AD and other dementias [6,7]. Recent studies have shown that plasma tau (pTau) phosphorelated at different sites (pTau181, pTau217, pTau231) might facilitate the differentiation between AD and other dementias [8]. The potential application for NfL is in the early detection of AD and the assessment of the course of the disease [6].

Therefore, blood based measurements of NfL (in serum or plasma) are promising, easily accessible biomarkers of neurodegeneration, and may thus contribute to define disease severity, quantify time to disease milestones and identify disease progression in AD [9].

It has been shown that AD patients have higher CSF and blood levels of NfL compared to cognitively unimpaired persons and persons with mild cognitive impairment [9,10,11,12]. High CSF and blood NfL levels in AD patients were found to correlate with worse performance in cognitive tests [9,10,12,13,14] and may predict cognitive decline [15,16,17]. Elevated NfL blood levels have been found even in the presymptomatic stage in familial [13,18,19] and sporadic AD [16,20], and have been suggested as a marker for tracking neurodegeneration in the different stages of AD [9,17,20,21].

We here contribute to the validation of sNfL as a neurodegeneration marker in AD by assessing the association between sNfL, cognitive functions, neuropsychiatric symptoms, and brain atrophy in the Prospective Dementia Registry-Austria (PRODEM-Austria). We hypothesized that baseline sNfL levels (1) relate to performance in cognitive tests and predict cognitive decline over time, (2) relate to neuropsychiatric symptoms and, (3) relate to cortical volume loss, particularly in AD signature regions. Our findings rely on a cohort of 237 patients with AD that were prospectively recruited in four university clinics in Austria and were followed over a mean period of 18 months using identical investigational protocols.

## 2. Materials and Methods

Participants: Data are from the prospective registry on dementia in Austria (PRODEM-Austria), a longitudinal, multi-centre cohort study supported by the Austrian Alzheimer Society. Inclusion criteria of PRODEM-Austria were diagnosis of dementia, no need for 24 h-care and availability of a caregiver who was able to provide information about the patients’ condition. Patients that could not sign the informed consent or had severe co-morbidities were excluded from the study.

For the present study, we used data collected from 23 June 2009 through 31 May 2017. All PRODEM participants had possible or probable AD according to the NINCDS-ADRDA criteria [22], the state-of-the art criteria at the time of patient recruitment, and serum NfL (sNfL) measurements. Exclusion criteria were other dementias than AD and cerebrovascular disease within six months before the sNfL measurement. Moreover, seven patients who were defined as outliers because of sNfL concentrations > 3 standard deviations of the study cohort mean were also excluded. The final study cohort consisted of 237 participants.

All participants underwent a detailed clinical assessment, cognitive testing, and laboratory evaluation at baseline, after six months, after one year and after two years. Of the included participants 221 (93.3%) had come to at least one follow-up visit. The mean follow-up time was 22 months. If participants had come to more than one follow-up visit, we used the one year follow-up visit.

A subgroup of 125 participants underwent MRI at baseline, and of those, 111 had at least one follow-up examination. The mean follow-up time in the MRI subgroup was 18 months. There was no significant difference in demographics and outcome measures between the MRI subgroup and the whole cohort.

Serum NfL: Serum samples were obtained by peripheral venipuncture and were frozen at −80 °C according to standard procedures [23]. The sNfL was quantified by an ultra-sensitive single molecule array (Simoa) on a Quanterix SR-X Analyzer (Quanterix, Billerica, MA, USA) using the commercially available NF-light^®^ assay. All measurements were done it duplicate to assure inter-assay precision by the calculating the coefficient of variation (CV). Results with CVs below 20% were accepted for statistical analysis.

Neuropsychological Tests: Global cognition was assessed using the mini mental state examination (MMSE) and the test battery “Consortium to Establish a Registry for Alzheimer’s Disease (CERAD)-Plus” [24]. The CERAD-Plus test included subtests for verbal memory and recognition, constructional praxis, figural memory, confrontational object naming, verbal fluency, and cognitive flexibility. The Neuropsychiatric Inventory (NPI) was used to assess frequency and severity of neuropsychiatric and behavioural symptoms. Besides the NPI total score, we defined three subgroups of neuropsychiatric symptoms based on Garre-Olmo et al. as follows: a “psychotic cluster” (including “delusions” and “hallucinations”), an “emotional cluster” (including “agitation”, “irritability”, “depression” and “anxiety”) and a “behavioural cluster” (including “euphoria”, “disinhibition”, “aberrant motor behaviour” and “apathy”). For each symptom we calculated a subscore by multiplying the frequency of the symptom by its severity. The NPI total score and the psychotic, emotional and behavioural cluster were calculated as sum of the corresponding subscores [25].

Magnetic Resonance Imaging: This was performed either on a 1.5 Tesla (T) whole body MR scanner (Avanto or SymphonyTim; Siemens Healthcare, Erlangen, Germany) or on a 3 T whole body MR scanner (TrioTim; Siemens Healthcare, Erlangen, Germany), with 1 mm isotropic resolution. Brain volumetry was assessed using the FreeSurfer image analysis suite (version 6.0, freely available online https://surfer.nmr.mgh.harvard.edu/, accessed: 11 March 2022). Technical details are described elsewhere [26,27]. We defined six regions of interest (ROI) of the volumetric data that have been shown to be sensitive for early AD. The ROI included the hippocampus, the parahippocampal cortex, the cuneus, the precuneus, the entorhinal cortex, and the inferior parietal cortex [28]. All regions were normalized for the estimated total intracranial volume and averaged between right and left sides.

Statistical analysis: Statistical analysis was performed using the IBM SPSS Statistics (Version 26.0, SPSS Inc., Chicago, IL, USA). Normally distributed continuous variables are presented using the mean and standard deviation (SD), nominal data is presented with absolute numbers and percentages. To assess normal distribution, we used the Shapiro–Wilk test and visual inspection of histograms. In case of non-normality, we used log-transformation to achieve normal distribution.

We used Pearson’s correlation and multiple linear regression to assess the relationship between (1) sNfL and cognitive function and (2) sNfL and cerebral atrophy. As it has been shown that sNfL increases with age, especially after around 60 years of age [29], we corrected for age in the linear regression analysis. In the analysis of the cognitive function, we further corrected for sex and education and in the analysis of the cerebral atrophy we corrected for sex. Because of statistical extreme values and non-linearity of the NPI results, we used Spearman’s correlation to assess the relationship between sNfL and neuropsychiatric and behavioural symptoms.

For the longitudinal analysis, we calculated the annualized absolute change of the cognitive and the MRI measures for each individual. All tests were 2-sided and a *p*-value (P) below 0.05 was considered as statistically significant. We used Bonferroni correction for multiple testing.

## 3. Results

Of 237 patients, 138 (58.2%) were female and 99 (41.8%) male. Their mean age was 75.4 years (SD = ±7.9). The mean MMSE at baseline was 22.46 (SD = ±3.77). The mean sNfL concentration was 28.19 pg/mL (SD = ±15.28) (Table 1). Baseline sNfL correlated with age (r = 0.378, *p* < 0.001), but not with years of education (r = 0.057, *p* = 0.382). There was no difference in baseline sNfl between men and women (mean sNfL 28.29 pg/mL ± 13.12 in women vs. 28.06 pg/mL ± 17.93 in men; *p* = 0.322).

### 3.1. sNfL and Cognition

The sNfL concentration was related to cognitive measures when adjusting for age, sex and education. As shown in Table 2, higher sNfL was significantly related to lower scores on the MMSE and to faster decline of MMSE scores.

Table 3 displays the cross-sectional and Table 4 the longitudinal results on the CERAD test battery. Patients with higher sNfL concentrations scored lower on the CERAD global test score. On CERAD subtests, associations were seen for language function (“Verbal Fluency”, “Boston Naming Test” and “S-Words”) and for constructional praxis. After correction for multiple comparisons, the association for the language subtests remained significant.

Longitudinally, no significant association was seen for the change in the CERAD total score. Yet, higher sNfL related to more rapid decline on subtests of language function, constructional praxis, and verbal memory. There was no significant association after correction for multiple comparisons.

### 3.2. sNfL and Neuropsychiatric Symptoms

At baseline, higher sNfL concentrations were seen in patients with higher NPI total scores. Looking at the different clusters individually, there was a significant correlation between sNfL and behavioural symptoms, while there existed no significant association with the psychotic and emotional NPI cluster. After correction for multiple comparisons, the correlations did not remain significant (Table 5). In the longitudinal analysis sNfL baseline levels were not significantly related to deterioration of neuropsychiatric symptoms (data not shown).

### 3.3. sNfL and MRI-Derived Global and Regional Brain Volumes

As can be seen in Table 6 higher sNfL levels were not associated with total cortical volume, but with smaller temporal lobe volume. The association between sNfL and AD signature regions was borderline significant. When considering the AD signature regions individually, we found negative associations between sNfL and the hippocampus, the entorhinal cortex and the parahippocampal cortex. After correction for multiple comparisons the associations with the parahippocampal cortex and the temporal lobe cortical volume remained significant.

Longitudinally, baseline sNfL levels were not related to annualized change in global or regional brain volumes (data not shown).

## 4. Discussion

In this prospective study in patients with AD, higher sNfL levels were associated with worse performance in cognitive tests, behavioural symptoms and smaller cerebral volumes of regions typically affected by AD pathology [28]. The sNfL levels at baseline predicted cognitive decline but not deterioration of neuropsychiatric behavioural symptoms and brain volume loss.

The observed association with cognitive performance is in line with previous results [9,11,12,14,15,30,31,32]. We found the strongest association between sNfL and cognitive functioning on subtests of language function. A somewhat weaker association was found between sNfL and constructional praxis. Like other studies we also found that baseline sNfL levels predicted more rapid future cognitive decline [15,16,21,33].

The sNfL was associated with the presence of neuropsychiatric symptoms measured by the NPI total score. Among the three symptomatic clusters of neuropsychiatric symptoms sNfL related only to the behavioural cluster. The correlation was modest, yet independent of age and sex. So far, only one other study has examined the relationship between CSF, NfL, and neuropsychiatric symptoms in AD patients and failed to observe a significant correlation [34]. It is unclear whether sNfL is directly related to neuropsychiatric symptomatology or the association is indirect because behavioural abnormalities indicate faster decline of cognitive functions [17,35]. Previously published data suggested an association between behavioural abnormalities and tau-associated pathology, as well as axonal degeneration in AD [36].

The brain regions that were strongest related to sNfL were the temporal lobes, hippocampal and parahippocampal cortex, as well as the entorhinal cortex. These findings replicate previous findings that reported associations of sNfL with lower volumes of hippocampus [9,31] and nonhippocampal AD signature regions [9,31,37]. We failed to find an association between baseline sNfL and progression of AD related atrophy. Presumably, the mean observational time of our study of 18 months was too short to detect change that exceeds measurement variability of automated longitudinal volumetric assessment [38,39].

The strengths of our study are the extensive diagnostic work-up including a detailed cognitive and neuropsychiatric behavioural assessment plus structural MRI scanning. One limitation of the study is that we used the NINCDS-ADRDA criteria for the diagnosis of AD [22]. These criteria were standard at the time of patient recruitment into the current investigation. Recently, the criteria have been revised, yet it was shown that the NINCDS-ADRDA criteria provide a diagnostic sensitivity and specificity of 81% and 70%, respectively, in clinicopathologic studies [40]. Further limitations were the relatively short follow-up time of 22 months in the whole cohort and 18 months in the MRI subgroup and the absence of a healthy control group.

## 5. Conclusions

In conclusion, our study supports the view that sNfL levels are associated with neurodegeneration and cognitive decline that is characteristic for AD. Recently, similar results have been reported for plasma p-tau181 [41] with the advantage that plasma p-tau181—other than sNfL—is considered to be AD-specific. Future studies will show if plasma p-tau181 will supplement sNfL as a neurodegeneration marker of AD, or whether the two blood biomarkers provide complementary information on the disease course and thus a combined use may be advantageous.

## Figures and Tables

**Table 1 medicina-58-00433-t001:** Demographics and Serum neurofilament light chain (NfL) (mean and SD).

	*N*	All Participants
Demographics		
Women (%)	237	138 (58.2)
Age (years)	237	75.4 ± 7.9
Education (years)	237	11 ± 2.5
Serum NfL		
sNfL baseline (pg/mL)	237	28.19 ± 15.28
sNfL annualized change (pg/mL)	221	1.79 ± 11.14

**Table 2 medicina-58-00433-t002:** Multiple linear regression analysis: sNfL baseline levels relate to mini mental state examination (MMSE) baseline score and change in MMSE score.

	*N*	Regression Coefficient (B)	Standard Error (SE)	*p* Value *
MMSE baseline	237	−3.014	1.293	0.021
MMSE change	235	−3.505	1.023	<0.001

* Corrected for age, sex, and education.

**Table 3 medicina-58-00433-t003:** Multiple linear regression analysis: significant associations between sNfL baseline level and baseline Consortium to Establish a Registry for Alzheimer’s Disease (CERAD) test score.

Test	*N*	B	SE	*p* Value *
CERAD Total Score	208	−10.084	2.999	<0.001
Verbal Fluency	216	−1.230	0.391	0.002 (0.022)
Boston Naming Test	215	−2.057	0.589	<0.001 (0.006)
Savings Constructional Praxis	214	−1.013	0.449	0.025 (0.275)
S-Words	213	−0.930	0.402	0.022 (0.242)

* Corrected for age, sex and education, correction for multiple comparisons in parenthesis.

**Table 4 medicina-58-00433-t004:** Multiple linear regression analysis: significant associations between sNfL baseline level and longitudinal change in CERAD subtests and CERAD Total Score.

Test	*N*	B	SE	*p* Value *
CERAD Total Score	160	−0.083	1.658	0.373
Wordlist Learning	197	−0.805	0.372	0.032 (0.352)
Constructional Praxis	195	−1.121	0.410	0.007 (0.077)
S-Words	194	−0.630	0.266	0.019 (0.209)

* Corrected for age, sex and education, correction for multiple comparisons in parenthesis.

**Table 5 medicina-58-00433-t005:** Partial correlation: sNfL baseline level and the Neuropsychiatric Inventory (NPI).

	*N*	Correlation Coefficient (r)	*p* Value *
NPI total score baseline	217	0.138	0.041
Psychotic cluster baseline	218	0.099	0.143 (0.429)
Emotional cluster baseline	217	0.075	0.266 (0.798)
Behavioural cluster baseline	218	0.135	0.045 (0.135)

* Corrected for age and sex, correction for multiple comparisons in parenthesis.

**Table 6 medicina-58-00433-t006:** Multiple linear regression analysis: sNfL baseline level and baseline volumetric measures.

	*N*	B	SE	*p* Value *
Total cortical volume	125	−0.012	0.006	0.069
Frontal lobe cortical volume	125	−0.004	0.005	0.427 (0.999)
Occipital lobe cortical volume	125	−0.001	0.001	0.550 (0.999)
Parietal lobe cortical volume	125	−0.003	0.003	0.317 (0.999)
Temporal lobe cortical volume	125	−0.012	0.003	0.001 (0.004)
Alzheimer’s dementia (AD) signature regions total volume	125	−0.004	0.002	0.050
Hippocampal volume	125	−0.001	0.000201	0.013 (0.078)
Cuneus volume	125	−0.000126	0.000200	0.528 (0.999)
Entorhinal cortex volume	125	−0.000308	0.000124	0.014 (0.084)
Inferior parietal cortex volume	125	−0.000458	0.000491	0.353 (0.999)
Parahippocampal cortex volume	125	−0.000316	0.000113	0.006 (0.036)
Precuneus volume	125	−0.000236	0.000356	0.509 (0.999)

* Corrected for age and sex, correction for multiple comparisons in parenthesis.

## Data Availability

The data sets are not publicly archived, but are available upon request to the corresponding author.

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
