# Peer review of "Serum NfL in Alzheimer Dementia: Results of the Prospective Dementia Registry Austria"

_medicina, 2022, doi:10.3390/medicina58030433_

Round 1

Reviewer 1 Report

This was a well designed and conducted study, with thought given to potential confounding variables in the designed models. These data support previous findings that sNfL is a predictor of future decline in AD using a separate cohort of patients to previously cited papers. sNfL was also linked to poorer cognitive performance at baseline.
Inclusion criteria were well described and methodologies adequately reported.  It would be interesting to see if, as the authors suggest, with longer follow up intervals greater changes would be observed in MRI measures of brain volume.

Typo in line 94. It should read "in duplicate".

Author Response

Response to Reviewer 1:

We would like to sincerely thank the reviewer for reading and assessing our work and for the constructive comments that helped improving this manuscript. In what follows we answered the comments in a point-by-point fashion. In the manuscript we indicated our revisions by the „Track Changes“ function in MS Word. Please see the attachment.

  1. It would be interesting to see if, as the authors suggest, with longer follow up intervals greater changes would be observed in MRI measures of brain volume.

Authors´ response: We thank the reviewer for this comment. We agree that longer follow up intervals would be required to assess whether sNfL can predict brain atrophy. In our study we had not planned a long term follow up. We might consider this in future studies.

  1. Typo in line 94. It should read "in duplicate".

Authors´ response: We corrected the typo. It now reads “All measurements were done in duplicate“.

We would like to thank the reviewer again for taking the time to review our manuscript.

Reviewer 2 Report

Daniela Kern and co-authors present the results of the search of associations between baseline serum Neurofilament light chain levels and cognitive, behavioral and volumetric findings in patients with Alzheimer's disese (AD). The results of the study are very important in the context of an AD diagnosis and prognostication of disease progression.

The authors mention several limitations of their study. Another limitation is the absence of the healthy control group.

There are several improvements which may make the manuscript better.

  1. AD is not the only disease which is characterized by presence of extracellular neurofilaments of tau protein. Another taupathies should be mentioned; besides, the parameters which allow to determine the exactly disease baised on serum neurofilament levels should be discussed.
  2. For easier reading and, more importantly, using the results in further studies the table containing absolute values of serum NfL levels should be added.

Author Response

Response to Reviewer 2:

We would like to sincerely thank the reviewer for reading and assessing our work and for the constructive comments that helped improving this manuscript. In what follows we answered the comments in a point-by-point fashion. In the manuscript we indicated our revisions by the „Track Changes“ function in MS Word. Please see the attachment.

  1. The authors mention several limitations of their study. Another limitation is the absence of the healthy control group.

Authors´ response: We agree that this is another limitation and have added this in our limitations of the manuscript (page 7, lines 241-243): “Further limitations are the relatively short follow-up time of 22 months in the whole cohort and 18 months in the MRI subgroup and the absence of a healthy control group.

  1. AD is not the only disease which is characterized by presence of extracellular neurofilaments of tau protein. Another taupathies should be mentioned; besides, the parameters which allow to determine the exactly disease based on serum neurofilament levels should be discussed.

Authors´ response: We thank the reviewer for this comment. We have added this in our Introduction (page 2, lines 47-54):

According to Bridel et al. the highest NfL levels are seen in HIV-associated dementia, frontotemporal dementia, vascular dementia, amyotrophic lateral sclerosis and atypical parkinsonian syndromes [5]. However, because of its low specificity for AD NfL is not suitable for the differentiation between AD and other dementias [6,7]. Recent studies have shown that plasma tau (pTau) phosphorelated at different sites (pTau181, pTau217, pTau231) might facilitate the differentation between AD and other dementias [8]. The potential application for NfL is in the early detection of AD and the assessment of the course of the disease [6].

  1. For easier reading and, more importantly, using the results in further studies the table containing absolute values of serum NfL levels should be added.

Authors´ response: We agree and provide a table in the Results (page 4, line 55). For easier reading we also added demographics to this table.

Table 1. Demographics and Serum NfL (mean and SD).

N

All participants

Demographics

Women (%)

Age (years)

Education (years)

237

237

237

138 (58.2)

75.4 ± 7.9

11 ± 2.5

Serum NfL

sNfL baseline (pg/ml)

sNfL annualized change (pg/ml)

237

221

28.19 ± 15.28

1.79 ± 11.14

We would like to thank the reviewer again for taking the time to review our manuscript.